# Association of Fish Oil Supplementation with Risk of Coronary Heart Disease in Individuals with Diabetes and Prediabetes: A Prospective Study in the UK Biobank

**DOI:** 10.3390/nu15143176

**Published:** 2023-07-17

**Authors:** Xiaohui Liu, Yin Li, Xuzhi Wan, Pan Zhuang, Yuqi Wu, Lange Zhang, Yang Ao, Jianxin Yao, Yu Zhang, Jingjing Jiao

**Affiliations:** 1Department of Endocrinology, The Second Affiliated Hospital, Zhejiang University School of Medicine, Hangzhou 310009, China; liuxiaohui@zju.edu.cn (X.L.); liyin-@zju.edu.cn (Y.L.); zhang_lange@126.com (L.Z.); aoyang990226@163.com (Y.A.); 2Department of Nutrition, School of Public Health, Zhejiang University School of Medicine, Hangzhou 310058, China; 3Zhejiang Key Laboratory for Agro-Food Processing, College of Biosystems Engineering and Food Science, Zhejiang University, Hangzhou 310058, China; xuzhiwan@zju.edu.cn (X.W.); panzhuang@zju.edu.cn (P.Z.); 22013038@zju.edu.cn (Y.W.); 22213038@zju.edu.cn (J.Y.); y_zhang@zju.edu.cn (Y.Z.)

**Keywords:** fish oil supplementation, coronary heart disease, diabetes, prediabetes, UK Biobank

## Abstract

This study aimed to explore the association between habitual intake of fish oil supplementation and the risk of developing CHD in patients with prediabetes and diabetes. Habitual use of fish oil was assessed by repeated questionnaires. Cox proportional hazard models were applied to calculate hazard ratios (HRs) and 95% confidence intervals (CIs). Over a median follow-up of 11.6 years, 4304 and 3294 CHD cases were documented among 47,663 individuals with prediabetes and 22,146 patients with diabetes in the UK Biobank, respectively. After multivariable adjustment, the HRs (95% CI) of CHD were 0.91 (0.85–0.98) and 0.87 (0.80–0.95) for individuals utilizing fish oil supplementation compared with non-users among the participants with prediabetes and diabetes, respectively. Furthermore, we identified an inverse relationship between fish oil use and CHD incidence, which was significantly mediated by serum C-reactive protein (CRP) levels in individuals with prediabetes and by very-low-density lipoprotein cholesterol (VLDL-C) in patients with diabetes at baseline. The inverse associations were consistent in the analyses stratified by potential confounders. In conclusion, the consumption of fish oil supplements was linked to decreased serum CRP and VLDL-C levels and subsequent CHD risk among adults with prediabetes and diabetes. Our findings highlight the important role of the habitual intake of fish oil supplements in preventing CHD in individuals with impaired glucose metabolism.

## 1. Introduction

Diabetes and prediabetes represent a fast-growing global healthcare burden (USD 966 billion) characterized by high morbidity rates (10.5% and 10.6%) and affecting 537 and 541 million people worldwide by 2021, respectively [1]. Epidemiological evidence has shown that patients with diabetes and prediabetes have a two-fold higher risk of developing coronary heart disease (CHD) compared with individuals with healthy blood glucose concentrations [2,3]. Moreover, people with prediabetes may experience CHD even without progression to diabetes [4]. Given that the prevention and control of hyperglycemia directly affect cardiovascular morbidity and mortality, immediate efforts are warranted to aid in the management of diabetes and prediabetes.

Fish oil supplements are widely consumed in developed countries [5] because their bioactive components docosahexaenoic acid (DHA) and eicosapentaenoic acid (EPA) are recommended for cardiovascular disease (CVD) prevention [6,7]. Although the American Heart Association recommends the intake of marine n-3 polyunsaturated fatty acids (PUFAs) for CVD prevention, limited evidence has been reported in participants with hyperglycemia, especially prediabetes. Furthermore, population-based studies have reported inconsistent findings [7]. Two large randomized controlled trials (RCTs) failed to detect a protective role of n-3 PUFA supplementation on CVD risk in participants with dysglycemia [8,9]. Nevertheless, the REDUCE-IT study found that 4 g/d icosapent ethyl reduced ischemic events, including cardiovascular death, compared with mineral oil placebo in individuals with CVD or risk factors, and the cardiovascular beneficial effect remained stable in the subgroup analysis of patients with diabetes [10]. The limited follow-up duration and newly diagnosed CVD cases in these clinical trials might have resulted in inadequate statistical power to detect significant effects. Moreover, the designated patients and well-controlled circumstances also limited the generalization of the findings of these trials to larger and more inclusive populations [11]. CHD is the most common disease in CVD among the general population and in patients with diabetes [2]. To date, few cohort studies have linked dietary consumption of n-3 PUFA, especially from marine products, with a reduced risk of developing CHD for patients with diabetes [12,13]. In addition, several studies have investigated the correlation between fish oil use and the risk of CVD [14,15,16], but they did not focus on individuals with impaired glucose metabolism, and different degrees of glucose metabolism disorder were not taken into consideration, which is important for mechanism exploration. Therefore, the association between long-term fish oil use and CHD prevention among adults with diabetes or prediabetes needs to be evaluated in real-world settings of large cohort studies.

To address the above concerns and fill the evidence gaps, we investigated the enduring association between habitual fish oil supplementation and the risk of CHD among 47,663 participants with prediabetes and 22,146 patients with diabetes in UK Biobank.

## 2. Methods

### 2.1. Study Population

UK Biobank enrolled over 500,000 individuals aged 37–73 years at 22 assessment centers located throughout the UK. From 2006 to 2010, individuals were invited to complete a detailed questionnaire, a face-to-face interview with physicians, and various anthropometric measurements, and they provided biological samples in assessment centers. All the individuals were required to provide an electronically signed informed consent form.

The definitions of diabetes and prediabetes are presented in Appendix A. Patients with diabetes were defined according to the algorithms of diabetes reported elsewhere [17]. In participants without diabetes, prediabetes meant an HbA1c level ranging from 5.7% to 6.4% (39–47 mmol/mol) [18]. A total of 31,384 patients with diabetes and 60,574 participants with prediabetes were initially enrolled in our study. We excluded adults with CVD or cancer at baseline, those without data on fish oil supplement use, and those who withdrew from UK Biobank. The final analysis included 22,146 patients with diabetes and 47,663 participants with prediabetes (Appendix A).

### 2.2. Assessment of Fish Oil Supplementation and Covariates

In various assessment centers, participants were asked whether they regularly took any supplements through a touch screen questionnaire. Participants had the option to select multiple choices from a designated list of supplements, which encompassed fish oil supplements. To reconfirm habitual fish oil supplementation, two additional repeated assessments were conducted on 22,146 and 47,663 participants in 2012–2013 and after 2014, respectively (Appendix A).

Information about other possible confounding variables was also collected by the baseline questionnaire, including sociodemographic factors, lifestyle factors, medical history, and drug use. Body mass index (BMI) was calculated as the weight (kg) divided by height squared (m^2^). The estimation of the Townsend deprivation index (TDI) served as a measure of socioeconomic status, considering factors such as lack of home ownership, absence of car ownership, unemployment, and household overcrowding [19]. The International Physical Activity Questionnaire (IPAQ) was applied to assess the metabolic equivalent of task (MET), serving as an indicator of the physical activity level [20]. Hypertension was defined as a systolic/diastolic blood pressure ≥ 140/90 mm Hg, a hospital diagnosis of hypertension, or the use of antihypertension drugs. High cholesterol was defined as a self-reported physician’s diagnosis or taking any cholesterol-lowering medications. The healthy diet score was calculated from 10 kinds of food according to the dietary recommendation for cardiometabolic health [21] (Appendix A).

A comprehensive array of biochemical markers was assessed in non-fasting blood samples collected at baseline (2006–2010), including serum levels of total cholesterol (TC), triglyceride (TG), low-density lipoprotein cholesterol (LDL-C), high-density lipoprotein cholesterol (HDL-C), apolipoprotein B (ApoB), apolipoprotein A (ApoA), random serum glucose, glycosylated hemoglobin (HbA1c), insulin-like growth factor-1 (IGF-1), C-reactive protein (CRP), and urate. The detailed methods of sample collection and treatment are described online [22]. Given that most serum CRP levels ranged between 0 and 1 with a right-skewed distribution, we applied a log (base 10) transformation in our analysis [23]. In addition, the sizes of lipoprotein subclasses, including very-low-density lipoprotein (VLDL) particles, LDL, IDL, and HDL particles, were measured by a high-throughput NMR-based platform. The details of the NMR platform, laboratory measurement, and quality control are available in the UKB online resource center [24].

### 2.3. Ascertainment of Coronary Heart Disease (CHD)

Incident CHD cases were identified through cumulative records of hospital inpatient data based on the International Classification of Diseases (ICD-9 and -10) and the Office of Population Censuses and Surveys Classification of Surgical Operations and Procedures. Additionally, self-reported data were obtained during the follow-up period to further enhance case identification (Appendix A). Hospital admission records were obtained by linking to the local databases, which were updated until 31 December 2020 [25]. Detailed information about the procedures is available online [https://biobank.ndph.ox.ac.uk/showcase/ukb/docs/HospitalEpisodeStatistics.pdf (accessed on 5 June 2023)]. The follow-up duration was calculated from baseline to the occurrence of CHD diagnosis, loss to follow-up, death, or the conclusion of the follow-up period (31 December 2020), whichever occurred first.

### 2.4. Statistical Analysis

Cox proportional hazards model was employed to estimate hazard ratios (HRs) and 95% confidence intervals (CIs) of CHD incidence associated with fish oil supplementation after checking for violation of the proportional hazard assumption. To control known and potential confounders, stepwise multivariable models were built. Model 1 adjusted age and sex. Then, model 2 further adjusted for race, assessment center, household income, TDI, BMI, smoking, alcohol drinking, physical activity, history of hypertension or hypercholesteremia, family history of CVD or diabetes, vitamin or mineral supplement use, and aspirin use. For patients with diabetes, diabetes duration was also adjusted in model 2. Model 3 further adjusted for the consumption frequency of oily fish, non-oily fish, processed meat, unprocessed red meat, poultry, vegetables, fruits, whole grains, refined grains, cheese, coffee, and sugar-sweetened beverages. Furthermore, the final multivariable model (model 4) additionally adjusted for the healthy diet score rather than specific foods and was based on model 2. In cases in which it was deemed necessary, missing data were coded using a designated missing indicator category. The healthy lifestyle score was also created on the basis of body shape (BMI < 30 kg/m^2^), physical activity (≥600 MET min/week), smoking (never), and healthy diet (yes) [26].

Given that the proportion of decreased CHD risk associated with fish oil supplementation might be mediated by intermediate biomarkers, such as inflammatory cytokines and blood lipids, we further conducted the mediation analysis to estimate the mediation proportion for fish oil supplementation [27]. A general linear model was used to evaluate the association between fish oil use and baseline biomarkers.

We conducted stratified analyses for the final model and examined potential interactions between fish oil supplement use and sex, age, BMI, TDI, smoking, alcohol drinking, physical activity, healthy diet score, healthy lifestyle score, history of hypertension, history of high cholesterol, family history of CVD, family history of diabetes, mineral supplement use, vitamin supplement use, aspirin use, non-oily fish intake, oily fish intake, and diabetes duration (for patients with diabetes). We investigated the potential interaction of fish oil supplementation with stratifying factors by adding the interaction term to the analyses. Sensitivity analyses were performed by further adjusting for lipid-lowering drugs, glucose-lowering drugs (for patients with diabetes), and sleeping patterns to assess the reliability and consistency of our findings. We also excluded individuals who had incident CHD within 2 years, individuals with missing covariate data, individuals who took other supplements, and individuals with type 1 diabetes (for patients with diabetes).

Statistical analyses were performed using SAS 9.4 (SAS Institute, Cary, NC, USA). *p* < 0.05 was considered significant for two-sided tests.

## 3. Results

### 3.1. Population Characteristics

Table 1 presents the characteristics of patients with prediabetes and diabetes at baseline according to fish oil supplementation. Of the 47,663 participants with prediabetes (26,816 women and 20,847 men) and 22,146 patients with diabetes (9273 women and 12,873 men), 15,936 (33.4%) and 6310 (28.5%) habitually consumed of fish oil supplements, respectively. Individuals who reported the use of fish oil supplements were slightly older, and more were women, white, not current smokers, and socioeconomically deprived. Compared with non-users, fish oil users demonstrated a higher likelihood of alcohol consumption, engagement in physical activity, a lower BMI, and adherence to a healthier diet. They also exhibited a lower likelihood of having a family history of diabetes and were more likely to have hypertension, hypercholesteremia, and a family history of CVD. They also showed a preference for using aspirin, other supplementation, and lipid-lowering medications. Among patients with diabetes, fish oil users were less likely to take glucose-lowering medication and had a shorter diabetes duration.

### 3.2. Fish Oil Supplementation and Risk of CHD

Among participants with prediabetes, a total of 4304 CHD events were reported over 11.1 years of follow-up (528,215 person-years). Among patients with diabetes, a total of 3294 CHD events were reported over an average of 10.6 years of follow-up (235,061 person-years). The regular use of fish oil supplements exhibited an inverse association with the risk of CHD in model 1, adjusting for age and sex (*p* = 0.004 for participants with prediabetes and *p* < 0.001 for patients with diabetes). After further adjustment of sociodemographic factors, other supplements, and drugs, the protective associations remained significant (model 2). The results remained stable after additional adjustment for dietary components, including the consumption of oily fish (model 3). Alternatively, after further adjustment for the healthy diet score instead of specific foods on the basis of model 2 (model 4), the HRs (95% CIs) were 0.91 (0.85–0.98) for participants with prediabetes (*p* = 0.009) and 0.87 (0.80–0.95) for patients with diabetes (*p* = 0.002) (Table 2).

### 3.3. Fish Oil Supplementation and Serum Biomarkers

Among the blood indicators, the observed negative association between fish oil supplementation and CHD incidence was mainly mediated by CRP, HDL-C, and IGF-1 in individuals with prediabetes, while HbA1c, VLDL-C, HDL-C, and ApoA primarily mediated the association in patients with diabetes (all *p <* 0.05). Small LDL-C significantly mediated the observed association (proportion mediated = 4.9%) (Table 3)**.** In secondary analyses, we further examined the cross-sectional relationship between fish oil supplementation and blood biomarkers at baseline. Fish oil supplementation was significantly associated with lower CRP levels after full adjustment for potential risk factors (*p* < 0.001 for prediabetes and *p* = 0.009 for diabetes). In addition, a positive relationship was identified between fish oil use and the serum levels of TC, LDL-C, HDL-C, and ApoB (all *p <* 0.05) among participants with prediabetes or diabetes. However, we verified positive associations between fish oil supplements and serum levels of TG and ApoA, and an inverse association between fish oil use and the serum level of HbA1c was only observed in patients with diabetes (all *p <* 0.05). For the participants with prediabetes, we only observed the association between fish oil use and a higher serum level of IGF-1 (*p <* 0.001) (Appendix A).

### 3.4. Subgroup and Sensitivity Analyses

We also performed subgroup analyses based on possible confounding factors for CHD (Figure 1). Similar results were observed across all the subgroups. Although no significant interaction effects were observed, marginal effects were observed for TDI (*p* for interaction = 0.080, for prediabetes), hypertension (*p* for interaction = 0.090, for diabetes), and aspirin use (*p* for interaction = 0.073, for diabetes).

In the sensitivity analyses, the results did not significantly change when further adjusted for lipid-lowering drugs, CRP levels, and sleep pattern score or after excluding incident CHD events within 2 years among individuals with diabetes and individuals with prediabetes. After excluding participants with missing covariate data (n = 31,618 for prediabetes and n = 4635 for diabetes), the protective association remained significant for individuals with prediabetes (*p* = 0.002) but disappeared for patients with diabetes (*p* = 0.117). Conversely, when we excluded participants who took other supplements at baseline (n = 31,169 for prediabetes and n = 15,267 for diabetes), the protective association remained significant for patients with diabetes (*p* = 0.008) but became nonsignificant for participants with prediabetes (*p* = 0.080). For patients with diabetes, the association between habitual fish oil supplementation and lower CHD risk was not materially changed when we further adjusted for glucose-lowering drugs or excluded participants with type 1 diabetes (Appendix A).

## 4. Discussion

In this prospective study, which included 3294 and 4304 incident cases of CHD among 22,146 patients with diabetes and 47,663 participants with prediabetes, habitual use of fish oil supplements was significantly linked with 9% and 13% lower incidence of CHD, independent of various confounding factors. The associations were mainly mediated by CRP in participants with prediabetes and VLDL-C in patients with diabetes.

A number of cohort studies have explored the relationship between n-3 PUFA intake and the risk of CHD [28], but few studies have assessed these associations in patients with diabetes. In the Nurses’ Health Study (from 1980 to 2014) and the Health Professionals Follow-Up Study (from 1986 to 2014), 646 CVD deaths were documented among 9053 women and 2211 men with diabetes. The consumption of marine n-3 PUFA was associated with a decreased risk of CVD mortality (HR_Q4vsQ1_ (95% CI): 0.69 (0.52–0.90), *P* for trend = 0.007) [29]. Regarding disease rather than mortality, the latest meta-analysis demonstrated that each additional serving of fish per week was related to an 8% lower risk of CHD but not myocardial infarction or stroke in patients with diabetes [30]. In our study, compared with the dietary intake of n-3 PUFAs, fish oil supplementation with increased levels and enhanced quality of DHA and EPA avoided interference from other food ingredients [31]. Meanwhile, we focused on CHD rather than CVD mortality in order to prevent the disease as early as possible. Another case–cohort study from the ADVANCE trial measured fatty acid levels in plasma samples collected among 3576 patients with diabetes at baseline. They found plasma n-3 PUFAs, especially plasma DHA, were negatively associated with macrovascular events (HR (95% CI): 0.87 (0.80–0.95)) [32]. Circulating concentrations of n-3 PUFAs are not subject to recall bias, which corroborates our results of patients with diabetes.

In contrast to the generally consistent results of cohort studies, results from RCTs have reported conflicting findings [33]. The findings of our study were consistent with those of a previous meta-analysis of 13 RCTs, indicating that n-3 PUFA supplements might exhibit modest efficacy in reducing the incidence and mortality of CHD [34]. Unfortunately, this study was not restricted to patients with diabetes, and the subgroup analyses of patients with diabetes were not available. Two large RCTs have explored the impact of n-3 PUFA supplementation on CVD mortality in participants with diabetes or with risk factors of diabetes. In the ORIGIN trial, 12,536 patients with dysglycemia received a daily 1 g capsule with 465 mg EPA and 375 mg DHA for at least 6 years. These results indicated that n-3 PUFA supplements did not decrease the risk of CVD death or other CVD cases, such as myocardial infarctions, stroke, revascularizations, heart failure, and angina [8]. In the ASCEND trial, which also used a dose of 1 g/day of fish oil supplementation with 460 mg EPA and 380 mg DHA in 15,480 patients with diabetes, a negative effect on the incidence of fatal myocardial infarction and total CHD was found after 7.4 years of intervention, although no effect was observed for major CVD outcomes [9]. Conversely, using the same dose as the ASCEND trial, n-3 PUFA supplementation had a marginal effect on major cardiovascular events and reduced the myocardial infarction risk by 60% in the subgroup analyses of 2728 diabetes patients in the VITAL trial [35]. Meanwhile, the REDUCE-IT study exhibited that 4 g/d of icosapent ethyl, a purified form of EPA, reduced CVD events and death in the subgroup analyses of 4787 diabetes patients; these results were similar to those of another trial with an intervention of 1.8 g/day EPA supplementation in 4565 individuals with type 2 diabetes or hyperglycemia over 4.6 years [10,36]. However, a 4 g/day carboxylic acid formulation of EPA and DHA did not show a cardiovascular benefit in the STRENGTH trial, which also enrolled participants with a high risk of CVD from several countries, as the REDUCE-IT study showed [37]. One possible explanation is the different doses of n-3 PUFA supplementation used in previous studies. Several meta-analyses have identified a dose-response correlation between marine n-3 PUFA supplementation and CVD risk, which indicates that low-dose n-3 PUFA supplements are not beneficial in reducing CVD events [34,38]. The choice of placebo may also be considered. Many RCTs have chosen olive oil as a placebo, which has anti-platelet and anti-inflammatory properties [39,40], and the use of olive oil might have mitigated the discrepancies between the groups, thus obscuring the effectiveness of n-3 PUFA supplementation [41]. Additionally, there were apprehensions regarding the potential adverse effects of using mineral oil as a placebo in the REDUCE-IT trial, as it is associated with a significant rise in C-reactive [10]. However, the Food and Drug Administration advisory committee declared that light mineral oil intake does not exert clinical effects on the endpoints of CVD incidence and mortality [42]. Additional concerns could be ascribed to the high intake of marine n-3 PUFAs at baseline. Supplementation with n-3 PUFA protected against major cardiovascular events and myocardial infarction in individuals with low intake fish intake but not in those with higher intake in the VITAL trial [35]. We also detected that the HR of CHD risk in patients with diabetes with higher oily fish consumption was higher than that in participants with lower consumption. Furthermore, the differential results observed between composite CVD endpoints and subtype outcomes indicate that the protective roles of marine n-3 PUFAs may not be uniform across all types of CVD [34]; thus, we focused on CHD rather than CVD based on previous findings.

Although the definitions of prediabetes are variable, more and more studies have suggested that participants with prediabetes are at higher risk of CVD regardless of the criteria, especially prediabetes defined by HbA1c levels [3,43]. Compared with diabetes, little is known about the beneficial effect of fish oil use in people with prediabetes, a condition with a much higher prevalence around the world [1]. In a prospective cohort study conducted in Japan, a high intake of n-3 PUFAs was not related to the risk of myocardial infarction among 1049 adults with prediabetes after 4.8 years of follow-up [13]. Regarding RCT trials, the JELIS study and the ORIGIN trial included participants with prediabetes but mixed them with patients with diabetes in the analyses, so they could not separately reveal the effect of n-3 PUFA supplementation on CVD risk in these two groups of people [8,36]. Using a larger sample size and an extended follow-up duration, this study detected that fish oil supplements were related to a lower risk of CHD among participants with prediabetes and patients with diabetes. The current findings highlighted a consistent and protective association of fish oil supplements with CHD risk at different stages of dysglycemia.

Multiple biological mechanisms might explain the negative relationship between fish oil supplements and CHD risk in participants with hyperglycemia, including the beneficial hypolipidemic and anti-inflammatory effects, which could act as a shield against the progression of CVD [44,45,46,47]. In participants with prediabetes, our results demonstrated an inverse association between fish oil supplementation and serum CRP levels, accounting for 5.9% of its relationship with CHD risk, aligning with the results of a previous meta-analysis of subjects with chronic diseases [48]. One possible mechanism through which the inverse association of n-3 PUFA intake with CRP level was ascribed to the regulation of inflammatory cytokines, including TNF-α and IL-6, by binding to peroxisome proliferator agonist receptors (PPAR) and inhibiting the stimulation of nuclear factor кB (NF-кB) [49,50]. In individuals with diabetes, fish oil supplementation might have a beneficial effect on lipid markers, especially VLDL-C, in agreement with another meta-analysis [51]. VLDL-C is capable of penetrating the arterial wall to enter and become trapped within the intima [52]. Marine n-3 PUFAs might downregulate the concentration of plasma VLDL-C by decreasing endogenous TG synthesis in the liver via the transcription of several nuclear receptors, such as peroxisome proliferator-activated receptor (PPAR)-α and sterol regulatory element binding proteins (SREBP) [53]. It is important to take LDL-C particle size into consideration in the management of patients with diabetes to prevent CHD. Although no significant mediation was found for total LDL-C, fish oil supplementation was inversely associated with small LDL-C, which mediated 3.9% of the protective association for CHD risk.

The major strength of our study is the large population size of patients with diabetes and participants with prediabetes, which provided sufficient statistical power and a large number of CHD cases to detect significant relationships by the joint and stratified analyses. Meanwhile, well-validated measures of biomarkers enabled us to investigate the relationships between the habitual intake of fish oil supplements and a series of serum biomarkers. Moreover, the detailed collection of information on socioeconomic characteristics, lifestyle factors, and covariates allowed for comprehensive adjustment and subgroup analyses. Some possible limitations should also be acknowledged. First, we used the data of self-reported fish oil use at baseline, which might not evaluate the longitudinal changes and thus may have caused misclassification. Nonetheless, we reconfirmed the status of fish oil use in the two repeated measurements (2012 to 2013 and 2014 and later) and found strong correlations among the three measurements, indicating the habitual use of fish oil supplementation. Second, we did not collect specific information about the formulation, dosage, and duration of fish oil use to further explore the dose–response relationship. Third, although we carefully controlled for various CHD risk factors in our models, residual or unmeasured confounding was still possible, such as measurement errors in the evaluation of diet. Fourth, the adults recruited in the UK Biobank study were mainly of white European descent, and thus caution is needed when generalizing our findings to other ethnic groups of populations. Fifth, we defined prediabetes according to the blood HbA1c level at baseline, but we did not take blood glucose into consideration, as the mean (SD) fasting time before blood sampling was only 3.9 (2.5) hours. In addition, even when including individuals with impaired glucose levels (5.6 to 6.9 mmol/L) in sensitivity analyses, the results remained stable. Finally, due to the observational nature, further confirmation is required to establish a causal relationship.

## 5. Conclusions

In summary, our findings suggest that regular intake of fish oil supplements is significantly related to a reduced risk of CHD in participants with diabetes and prediabetes. The inverse relationship between fish oil supplementation and CHD risk was significantly mediated by serum CRP in individuals with prediabetes and VLDL-C in patients with diabetes at baseline. These results provide epidemiological evidence supporting the use of fish oil supplements for the primary prevention of CHD in patients with diabetes and individuals with prediabetes. Additional studies are required to explore the optimal dosage and duration of fish oil supplementation to prevent CVD.

## Figures and Tables

**Figure 1 nutrients-15-03176-f001:**
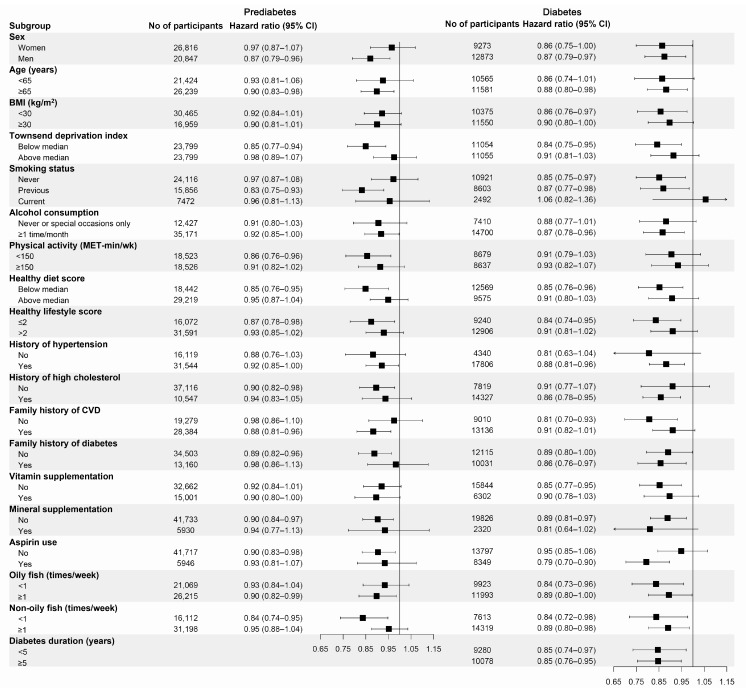
Subgroup analyses for the association of fish oil supplementation with the risk of CHD stratified by confounding factors. BMI, body mass index; MET, metabolic equivalent of task. Forest plots show the multivariable HRs of CHD associated with fish oil use in subgroups. HRs were adjusted for age (continuous), sex (male or female), race (white or non-white), assessment center (22 categories), BMI (in kg/m^2^; <18.5, 18.5 to 25, 25 to 30, 30 to 35, ≥35, or missing), Townsend deprivation index (tertiles), household income (<GBP 18,000, GBP 18,000–GBP 30,999, GBP 31,000–GBP 51,999, GBP 52,000–GBP 100,000, >GBP 100,000, or missing), smoking status (never, former, current, or missing), alcohol drinking (never or special occasions only, 1 or 2 times/week, 3 or 4 times/week, ≥5 times/week, or missing), physical activity (in MET-h/wk; tertiles), history of hypertension (yes or no), history of high cholesterol (yes or no), family history of cardiovascular diseases (yes or no), family history of diabetes (yes or no), vitamin supplement use (yes or no), mineral supplement use (yes or no), aspirin use (yes or no), and healthy diet score (tertiles). Results were further adjusted for diabetes duration for patients with diabetes (years; <5, 5 to 10, ≥10).

**Table 1 nutrients-15-03176-t001:** Basic characteristics of individuals with prediabetes and diabetes stratified by fish oil use.

Characteristics	Prediabetes				Diabetes			
	Overall	Fish Oil Non-Users	Fish Oil Users	*p*-Value	Overall	Fish Oil Non-Users	Fish Oil Users	*p*-Value
	(n = 47,663)	(n = 31,727)	(n = 15,936)		(n = 22,146)	(n = 15,836)	(n = 6310)	
Male/n (%)	43.7	45.7	39.8	<0.001	58.1	58.6	57.0	0.038
Age (year)	59.0 ± 7.1	58.1 ± 7.4	60.7 ± 6.3	<0.001	58.4 ± 7.6	57.7 ± 7.7	60.1 ± 6.9	<0.001
Race/n (%)				<0.001				<0.001
White	89.2	88.6	90.6		85.6	85.0	87.3	
Nonwhite/mixed	10.3	11.0	9.0		13.8	14.5	12.3	
BMI (kg/m^2^)	28.9 ± 5.3	29.1 ± 5.4	28.5 ± 5.0	<0.001	31.2 ± 5.9	31.4 ± 6.0	30.9 ± 5.7	<0.001
Household income (GBP)				<0.001				0.016
<18,000 *	24.2	23.9	24.6		27.5	27.6	27.1	
18,000 to 30,999	23.6	22.9	25.1		23.0	22.6	24.1	
31,000 to 51,999	20.0	20.4	19.2		18.1	17.9	18.5	
52,000 to 100,000	12.1	12.8	10.6		11.5	11.8	10.8	
>100,000	2.8	3.1	2.3		2.5	2.7	2.2	
Townsend deprivation index	−1.0 ± 3.2	−0.9 ± 3.3	−1.3 ± 3.1	<0.001	−0.5 ± 3.4	−0.4 ± 3.4	−0.8 ± 3.3	<0.001
Smoking status/n (%)				<0.001				<0.001
Never	50.6	50.4	51.1		49.3	50.3	47.0	
Previous	33.3	31.7	36.4		38.9	37.2	42.9	
Current	15.7	17.5	12.2		11.3	12.0	9.3	
Alcohol consumption/n (%)				<0.001				<0.001
<1 times/week	38.5	39.6	36.1		45.9	47.4	42.0	
1 or 2 times/week	25.2	24.9	25.6		23.4	23.0	24.5	
3 or 4 times/week	19.1	18.4	20.4		15.8	15.1	17.4	
≥5 times/week	17.2	16.9	17.8		14.8	14.3	16.0	
Physical activity (MET-h/wk)	44.8 ± 47.1	43.3 ± 46.9	47.6 ± 47.3	<0.001	38.1 ± 43.3	36.3 ± 42.6	42.6 ± 44.8	<0.001
History of hypertension/n (%)	66.2	65.5	67.6	<0.001	80.4	80.0	81.5	0.008
History of high cholesterol/n (%)	22.1	21.0	24.4	<0.001	64.7	63.9	66.6	<0.001
Family history of CVD/n (%)	59.6	58.9	60.8	<0.001	59.3	58.8	60.6	0.018
Family history of diabetes/n (%)	27.6	27.9	27.0	0.032	45.3	45.6	44.6	0.197
Aspirin use/n (%)	12.5	11.3	14.8	<0.001	37.7	35.9	42.2	<0.001
Lipid-lowering drug/n (%)	21.0	20.1	22.9	<0.001	65.5	64.9	66.9	0.006
Glucose-lowering medication/n (%)								
Oral	-	-	-	-	50.6	51.2	49.3	0.010
Insulin	-	-	-	-	17.2	17.8	15.5	<0.001
Vitamin supplementation/n (%)	31.5	19.6	55.2	<0.001	28.5	18.2	54.1	<0.001
Mineral supplementation/n (%)	12.4	8.3	20.7	<0.001	10.5	7.7	17.5	<0.001
Dietary consumption								
Oily fish (times/week)				<0.001				<0.001
<1	44.2	48.3	36.1		44.8	48.3	36.0	
1	37.1	35.2	40.9		34.9	33.3	38.9	
≥2	17.9	15.6	22.5		19.3	17.2	24.4	
Non-oily fish (times/week)				<0.001				<0.001
<1	33.8	36.3	28.9		34.4	36.3	29.5	
1	49.4	47.9	52.4		47.8	46.5	51.3	
≥2	16.0	15.0	18.2		16.8	16.2	18.5	
Poultry (times/week)				<0.001				0.026
<2	53.5	53.7	53.1		51.1	51.5	50.2	
2–4	44.0	43.6	44.8		45.4	44.8	46.7	
>4	2.3	2.5	1.9		3.1	3.2	2.8	
Processed meat (times/week)				<0.001				<0.001
<1	37.9	36.7	40.2		33.4	32.8	35.1	
1	29.0	28.6	29.7		28.9	28.7	29.3	
≥2	32.8	34.3	29.8		37.2	38.1	35.1	
Unprocessed red meat (times/week)				<0.001				0.563
<2.0	47.8	47.5	48.4		45.2	45.4	44.8	
2.0–4.0	42.6	42.5	42.9		43.8	43.6	44.5	
>4.0	9.4	9.8	8.6		10.7	10.8	10.4	
Vegetables (servings/day)				<0.001				<0.001
<1.0	18.8	20.7	15.1		19.7	21.4	15.6	
1.0–2.9	71.6	69.8	75.3		69.1	67.9	72.1	
≥3.0	8.7	8.6	9.0		10.1	9.6	11.3	
Fruits (servings/day)				<0.001				<0.001
<2.0	36.7	40.3	29.5		30.7	32.8	25.4	
2.0–3.9	46.2	44.4	49.8		48.1	47.2	50.3	
≥4.0	16.8	14.9	20.5		20.8	19.6	24.0	
Whole grains (servings/day)				<0.001				<0.001
<1.0	46.3	50.0	38.9		43.9	47.0	36.1	
1.0–2.9	40.6	37.6	46.6		39.3	37.2	44.6	
≥3.0	12.0	11.3	13.5		15.7	14.7	18.3	
Refined grains (servings/day)				<0.001				<0.001
<1.0	54.5	51.7	60.0		53.3	51.1	58.9	
1.0–2.9	33.5	35.3	30.1		33.3	34.4	30.6	
≥3.0	10.9	11.9	8.9		12.3	13.3	9.6	
Cheese (pieces/day)				0.013				0.101
<2	41.3	41.1	41.7		46.5	47.0	45.5	
2–4	43.7	43.6	44.1		39.4	38.9	40.6	
>4	12.3	12.7	11.6		10.0	10.1	9.7	
Coffee (cups/day)				<0.001				<0.001
<1	30.4	31.2	28.9		31.4	32.1	29.5	
1–2	37.8	35.8	41.8		35.6	34.1	39.3	
≥3	31.5	32.6	29.1		32.4	33.2	30.6	
Sugar-sweetened beverages consumer/n (%)	83.3	84.5	81.0	<0.001	57.3	58.5	54.2	<0.001
Healthy diet score	3.0 ± 1.4	2.8 ± 1.4	3.2 ± 1.4	<0.001	3.3 ± 1.5	3.2 ± 1.5	3.5 ± 1.5	<0.001
Diabetes duration (year)								0.126
<5	-	-	-	-	41.9	41.7	42.4	
5–10	-	-	-		22.4	22.2	23.0	
≥10	-	-	-		23.1	23.3	22.7	

BMI, body mass index; MET, metabolic equivalent. Values are mean ± SD or percentages unless stated otherwise. * GBP 1.00 = USD 1.30, EUR 1.20.

**Table 2 nutrients-15-03176-t002:** Hazard ratios (HRs) and 95% confidence intervals (CIs) for the incidence of CHD based on fish oil use in participants with prediabetes and patients with diabetes.

	Fish Oil Supplementation		*p*-Value
	Non-Users	Users	
**Prediabetes**			
n (%)	31,727 (66.6)	15,936 (33.4)	
Cases of CHD (%)	2871 (9.1)	1433 (9.0)	
Person-years	351,221	176,993	
Model 1	1 [Ref.]	0.91 (0.85–0.97)	0.004
Model 2	1 [Ref.]	0.91 (0.85–0.97)	0.006
Model 3	1 [Ref.]	0.92 (0.86–0.99)	0.018
Model 4	1 [Ref.]	0.91 (0.85–0.98)	0.009
**Diabetes**			
n (%)	15,836 (71.5)	6310 (28.5)	
Case of CHD (%)	2393 (15.1)	901 (14.3)	
Person-years	167,867	67,194	
Model 1	1 [Ref.]	0.85 (0.79–0.92)	<0.001
Model 2	1 [Ref.]	0.87 (0.80–0.95)	0.001
Model 3	1 [Ref.]	0.88 (0.81–0.96)	0.003
Model 4	1 [Ref.]	0.87 (0.80–0.95)	0.001

Model 1: adjustment for age (continuous) and sex (male or female). Model 2: model 1 + race (white or non-white), assessment center (22 categories), BMI (in kg/m^2^; <18.5, 18.5 to 25, 25 to 30, 30 to 35, ≥35, or missing), Townsend deprivation index (tertiles), household income (<GBP 18,000, GBP 18,000–GBP 30,999, GBP 31,000–GBP 51,999, GBP 52,000–GBP 100,000, >GBP 100,000, or missing), smoking status (never, former, current, or missing), alcohol drinking (never or special occasions only, 1 or 2 times/week, 3 or 4 times/week, ≥5 times/week, or missing), physical activity (in MET-h/wk; tertiles), hypertension (yes or no), hypercholesterolemia (yes or no), family history of cardiovascular diseases (yes or no), family history of diabetes (yes or no), vitamin supplement use (yes or no), mineral supplement use (yes or no), and aspirin use (yes or no). Further adjusted diabetes duration for patients with diabetes (years; <5, 5 to 10, ≥10). Model 3: model 2 + oily fish (<1, 1, or ≥2 times/week), non-oily fish (<1, 1, or ≥2 times/week), poultry (<2, 2–4, >4 times/week), vegetables (<1.0, 1.0–2.9, ≥3.0 servings/day), fruits (<2.0, 2.0–3.9, ≥4.0 servings/day), whole grains (<1.0, 1.0–2.9, ≥3.0 servings/day), refined grains (<1.0, 1.0–2.9, ≥3.0 servings/day), chees (<2, 2–4, >4 times/week), coffee (<1, 1–2, ≥3 cups/day), and sugar-sweetened beverages (yes or no). Model 4: model 2 + healthy diet score (tertiles).

**Table 3 nutrients-15-03176-t003:** Mediation analysis between fish oil use and blood biomarkers for CHD risk.

Mediator		Prediabetes			Diabetes	
	Cases/N	Proportion (%) of Effect Due to Mediation (95% CI)	*p*-Value	Cases/N	Proportion (%) of Effect Due to Mediation (95% CI)	*p*-Value
Glucose	3770/41,648	NA	NA	2851/19,032	NA	NA
HbA1c	4304/47,663	NA	NA	3099/20,726	2.5% (0.7–8.6%)	0.034
CRP *	4085/45,309	5.9% (1.9–16.9%)	0.003	3097/20,714	2.1% (0.3–11.6%)	0.125
TG	4095/45,391	NA	NA	3103/20,740	NA	NA
TC	4096/45,426	NA	NA	3109/20,775	NA	NA
VLDL-C	992/11,366	NA	NA	796/5226	6.1% (1.5–22.5%)	0.033
Large VLDL-C	992/11,366	NA	NA	796/5226	3.7% (0.7–18.3%)	0.084
Medium VLDL-C	992/11,366	NA	NA	796/5226	3.9% (0.7–18.9%)	0.083
Small VLDL-C	992/11,366	NA	NA	796/5226	5.3% (1.1–21.9%)	0.058
Very large VLDL-C	992/11,366	NA	NA	796/5226	4.9% (1.1–19.4%)	0.044
Very small VLDL-C	992/11,366	NA	NA	796/5226	3.8% (0.6–20.6%)	0.111
LDL-C	4091/45,336	NA	NA	3101/20,707	NA	NA
Large LDL-C	992/11,366	NA	NA	796/5226	2.6% (0.4–14.7%)	0.104
Medium LDL-C	992/11,366	NA	NA	796/5226	3.9% (0.8–17.2%)	0.058
Small LDL-C	992/11,366	NA	NA	796/5226	4.9% (1.1–19.2%)	0.039
IDL-C	992/11,366	NA	NA	796/5226	2.3% (0.3–15.4%)	0.135
HDL-C	3773/41,665	3.8% (0.9–15.6%)	0.033	2852/19,039	3.1% (1.0–9.6%)	0.018
Large HDL-C	992/11,366	NA	NA	796/5226	NA	NA
Medium HDL-C	992/11,366	NA	NA	796/5226	1.6% (0.2–14.9%)	0.186
Small HDL-C	992/11,366	NA	NA	796/5226	NA	NA
Very large HDL-C	992/11,366	NA	NA	796/5226	NA	NA
Apo A	3760/41,516	2.8% (0.4–17.0%)	0.116	2844/18,996	3.1% (1.0–9.1%)	0.013
Apo B	4070/45,148	NA	NA	3084/20,544	NA	NA
IGF-1	4077/45,173	3.2% (1.0–10.3%)	0.013	3092/20,644	NA	NA

CRP, C-reactive protein; TG, triglyceride; TC, total cholesterol; HDL, high-density lipoprotein; LDL, low-density lipoprotein. * log (units + 1) of CRP values were used. Results were adjusted for age (continuous), sex (male or female), race (white or non-white), assessment center (22 categories), BMI (in kg/m^2^; <18.5, 18.5 to 25, 25 to 30, 30 to 35, ≥35, or missing), Townsend deprivation index (tertiles), household income (<GBP 18,000, GBP 18,000–GBP 30,999, GBP 31,000–GBP 51,999, GBP 52,000–GBP 100,000, >GBP 100,000, or missing), smoking status (never, former, current, or missing), alcohol drinking (never or special occasions only, 1 or 2 times/week, 3 or 4 times/week, ≥5 times/week, or missing), physical activity (in MET-h/wk; tertiles), history of hypertension (yes or no), history of high cholesterol (yes or no), family history of cardiovascular diseases (yes or no), family history of diabetes (yes or no), vitamin supplement use (yes or no), mineral supplement use (yes or no), aspirin use (yes or no), and healthy diet score (tertiles). Results were further adjusted for diabetes duration for patients with diabetes (years; <5, 5 to 10, ≥10).

## Data Availability

The data used in this current study are available from the UK Biobank data resources [https://www.ukbiobank.ac.uk/ (accessed on 5 June 2023)].

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
