# Peer review of "Association of Fish Oil Supplementation with Risk of Coronary Heart Disease in Individuals with Diabetes and Prediabetes: A Prospective Study in the UK Biobank"

_nutrients, 2023, doi:10.3390/nu15143176_

Round 1
Reviewer 1 Report
This very well written manuscript uses data from UKBioBank to explore the association between habitual use of fish oil supplements and incident CHD in people with diabetes or prediabetes. Strengths are sample size, duration of follow up and information about confounders. The research is conducted to a very good standards. The manuscript is well written and tables and figures show the data really well. It is found that use of fish oil supplements lowers risk of CHD in both diabetics and prediabetics. The effect holds up through four different statistical models that take various confounders into account. The effect in diabetics may involve CRP while in prediabetics it may involve VLDL-C. The discussion is good and deals well with context and with strengths and limitations.
Points for attention:
1. Title and Abstract should both mention UK BioBank.
2. Page 1. glycemic levels -> blood glucose concentrations
3. high purity of -> bioactive components
4. Page 2. REDUC-IT -> REDUCE-IT
5. ethyl had reduced -> ethyl reduced
6. It is not clear what "ideal and controlled concentrations" means
7. The sentence "CHD is the most popular ..." does not make sense
8. Please write PUFA in full
9. not took -> not taken
10. Section 3.3, line 5. Delete "was"
11. Page 11. TNF and Il-6 are not "transcription factors". Please rewrite this sentence.
Some changes required - see above
Author Response
- Title and Abstract should both mention UK BioBank.
Answer: Done. We have revised the title.
Title:
Association of Fish Oil Supplementation with Risk of Coronary Heart Disease in Diabetes and Prediabetes Individuals: A Prospective Study in the UK Biobank
Abstract, Lines 4-6:
Over a median follow-up of 11.6 years, 4304 and 3294 CHD cases were documented among 47,663 prediabetes and 22,146 diabetes patients in the UK Biobank.
- Page 1. glycemic levels -> blood glucose concentrations
Answer: Done.
Introduction, Paragraph 1, Line 6:
Epidemiological evidence has shown that not only diabetic patients but also participants with prediabetes have a two-fold higher risk of coronary heart disease (CHD) than those with healthy blood glucose concentrations [2,3].
- high purity of -> bioactive components
Answer: Done.
Introduction, Paragraph 2, Line 2:
Fish oil supplements are widely consumed in developed countries [5] because their bioactive components of docosahexaenoic acid (DHA) and eicosapentaenoic acid (EPA) were recommended for cardiovascular disease (CVD) prevention [6,7].
- Page 2. REDUC-IT -> REDUCE-IT
Answer: Done.
Introduction, Paragraph 2, Line 9:
Nevertheless, the REDUCE‐IT study found that 4 g/d icosapent ethyl reduced ischemic events, including cardiovascular death, as compared to mineral oil placebo among individuals with CVD or risk factors, and the cardiovascular beneficial effect remained stable in the subgroup analysis of diabetic patients [10].
- ethyl had reduced -> ethyl reduced
Answer: Done.
Introduction, Paragraph 2, Line 10:
Nevertheless, the REDUCE‐IT study found that 4 g/d icosapent ethyl reduced ischemic events, including cardiovascular death, as compared to mineral oil placebo among individuals with CVD or risk factors, and the cardiovascular beneficial effect remained stable in the subgroup analysis of diabetic patients [10].
- It is not clear what "ideal and controlled concentrations" means
Answer: Done. Controlled circumstances are environments controlled for some confounders that differ from the realities. We have revised the sentence to clarify the means.
Introduction, Paragraph 2, Line 15:
Moreover, the designated patients, well controlled circumstances also limit the generalization of findings from these trials to larger and more inclusive populations [11].
- The sentence "CHD is the most popular ..." does not make sense
Answer: Done. We have revised the phrase.
Introduction, Paragraph 2, Line 16:
CHD is the most common disease in CVD than other subtypes among general population and so does in diabetic patients
- Please write PUFA in full
Answer: When the word first appears in the manuscript, we spelled it in full. After that, we used the abbreviation as other studies did [1-3].
Introduction, Paragraph 2, Line 3-7:
Although the American Heart Association recommends the intake of marine n-3 polyunsaturated fatty acids (PUFAs) for the CVD prevention, limited evidence has been investigated from participants with hyperglycemia, especially prediabetes, and furthermore, population-based studies have reported inconsistent findings [7].
- not took -> not taken
Answer: Done.
Introduction, Paragraph 2, Line 22:
In addition, several studies investigated the correlation between fish oil use and the risk of CVD [14-16], they didn’t focus on individuals with impaired glucose metabolism and different degree of glucose metabolism disorder was not taken into consideration which was important for mechanism exploration
- Section 3.3, line 5. Delete "was"
Answer: Done.
Section 3.3, Paragraph 1, Line 4:
Small LDL-C significantly mediated the observed association (proportion mediat-ed=4.9%) (Table 3).
- Page 11. TNF and Il-6 are not "transcription factors". Please rewrite this sentence.
Answer: Done. We have revised the sentence.
Discussion, Paragraph 5, Line 9:
One possible mechanism through which the inverse association of n-3 PUFA intake with CRP level was ascribed to the regulation of inflammatory cytokines, including TNF-α and IL-6, by binding to peroxisome proliferator agonist receptors (PPAR) and inhibiting the stimulation of nuclear factor кB (NF-кB) [49,50].
Reviewer 2 Report
This is a well analyzed paper.
My only substantive concern is "habitual users" of supplements. What percent of those indicating use at baseline were using fish oil on the 2nd and 3rd surveys. Please provide a number. If not that high, consider looking at the subset with positive answers on 2 of 3 or 3 of 3 surveys.
There are minor English issues:
INTRO - REDUCE-IT.....Popular should be common....was not taken into consideration
Section 3.4 - continues should be continuous ..... Oppositely should be on the contrary
DISC - " we focused on fish oil...in order to prevent this disease early on. "Rewrite please. Hard to understand.
A good paper
Have an English editor review the paper. There are several small mistakes
Author Response
My only substantive concern is "habitual users" of supplements. What percent of those indicating use at baseline were using fish oil on the 2nd and 3rd surveys. Please provide a number. If not that high, consider looking at the subset with positive answers on 2 of 3 or 3 of 3 surveys.
Answer: We appreciate the reviewer’s suggestion. As shown in Supplemental Table 1, though some participants were invited to the assessments to finish a repeat touchscreen questionnaire including the question about fish oil supplementation intake, only 4801 participants with prediabetes (1705 participants in 2nd and 3703 participants in 3rd survey) and 1911 diabetic patients (738 participants in 2nd and 1416 participants in 3rd survey) had ≥ 2 times of data regarding fish oil use. The number of repeated measurements accounted for only 8-10% of the baseline population. Thus, it did not make sense to conduct a subgroup analysis among participants with ≥ 2 times of data regarding fish oil use.
Discussion, Paragraph 6, Line 8-13:
First, we used the data of self-reported fish oil use at baseline, which might not evaluate the longitudinal changes and thus cause misclassification. Nonetheless, we recon-firmed the status of fish oil use in the two repeated measurements (2012 to 2013 and 2014 and later) and found strong correlations among the three measurements, indicating their habitual use of fish oil supplementation.
There are minor English issues:
INTRO - REDUCE-IT.....Popular should be common....was not taken into consideration
Answer: Done.
Introduction, Paragraph 2, Line 10:
Nevertheless, the REDUCE‐IT study found that 4 g/d icosapent ethyl reduced ischemic events, including cardiovascular death, as compared to mineral oil placebo among individuals with CVD or risk factors, and the cardiovascular beneficial effect remained stable in the subgroup analysis of diabetic patients [10].
Introduction, Paragraph 2, Line 16:
CHD is the most common disease in CVD than other subtypes among general population and so does in diabetic patients [2].
Introduction, Paragraph 2, Line 22:
In addition, several studies investigated the correlation between fish oil use and the risk of CVD [14-16], they didn’t focus on individuals with impaired glucose metabolism and different degree of glucose metabolism disorder was not taken into consideration which was important for mechanism exploration
Section 3.4 - continues should be continuous.....Oppositely should be on the contrary
Answer: Done.
Table 2:
Model 1: results were adjusted for age (continuous), sex (male or female).
Table 3:
Results were adjusted for age (continuous)……
Figure 1:
HRs were adjusted for age (continuous) ……
Section 3.4, Paragraph 2, Line 6:
On the contrary, when we further excluded participants who took other supplements at baseline (n = 31,169 for prediabetes and n = 15,267 for diabetes), the protective association remained significant for diabetic patients (P = 0.008) but became nonsignificant for prediabetes participants (P = 0.080).
DISC - " we focused on fish oil...in order to prevent this disease early on. "Rewrite please. Hard to understand.
Answer: Done. We have rewritten the sentence.
Discussion, Paragraph 2, Line 9-12:
In our study, compared with dietary intake of n-3 PUFAs, fish oil supplementation with increased levels and enhanced quality of DHA and EPA avoided interference from other food ingredients [31]. Meanwhile, we focused on CHD rather than CVD mortality in order to prevent disease as early as possible.
Comments on the Quality of English Language
A good paper
Have an English editor review the paper. There are several small mistakes
Answer: Thank you for your suggestion. We have thoroughly checked the English.
Reviewer 3 Report
Liu et al. aimed to examine the association of habitual fish oil supplementation with incident CHD among patients with prediabetes and diabetes.
The study is well-written and shows many data about the effects of fish oil supplementation.
The authors do not mention in the Methods section how many women and how many men were enrolled in the study.
In the Methods section, the authors mention that the age of the enrolled participants were 37-73 years old; however, its significance is not presence in the study. The biomarkers of CHD risk, so the manifestation of diabetes is largely determined by age. Please, add some information (as textual explanation) to the Discussion about the effect of age groups (and fish oil supplementation) on diabetic condition.
Author Response
The authors do not mention in the Methods section how many women and how many men were enrolled in the study.
Answer: Done. We have now added the specific number of women and men in the results. Meanwhile, the percentage of male is shown in Table 1.
Section 3.1, Paragraph 1, Line 3-4:
Table 1 presents the baseline characteristics of participants with prediabetes and diabetes according to fish oil supplementation. Of the 47,663 participants with prediabetes (26,816 women and 20,847 men) and 22,146 diabetic patients (9273 women and 12,873 men), 15,936 (33.4%) and 6310 (28.5%) reported habitual use of fish oil supplements, respectively.
In the Methods section, the authors mention that the age of the enrolled participants were 37-73 years old; however, its significance is not presence in the study. The biomarkers of CHD risk, so the manifestation of diabetes is largely determined by age. Please, add some information (as textual explanation) to the Discussion about the effect of age groups (and fish oil supplementation) on diabetic condition.
Answer: We appreciate the reviewer’s comment. We agree with the important role of age in the development of CHD and we have also adjusted age as a continuous factor in the final model that contributed most to the models (χ2 = 185.1 for diabetes and χ2 = 247.8 for prediabetes). However, fish oil users were significantly older than non-users (Table 1). It is reasonable that elderly individuals exhibit a higher propensity for consuming dietary supplements. Thus, the protective association between fish oil supplementation and incident CHD risk was not mediated by age in our study.
Round 2
Reviewer 3 Report
I accept the revised version of the MS.
Author Response
Thanks!